# The Spatiotemporal Relationship between Landslides and Mechanisms at the Heifangtai Terrace, Northwest China

Tianfeng Gu [1], Jiading Wang [1,*], Henry Lin [2], Qiang Xue [2], Bin Sun [1], Jiaxu Kong [1], Jiaxing Sun [1], Chenxing Wang [1], Fanchen Zhang [1] and Xiao Wang [1]

1   State Key Laboratory of Continental Dynamics, Department of Geology, Northwest University, Xi'an 710069, China; gutf@nwu.edu.cn (T.G.); andy_lauzm@163.com (B.S.); 18829014384@163.com (J.K.); sunjx368@163.com (J.S.); 2019026006@chd.edu.cn (C.W.); zhangfanchen@163.com (F.Z.); wxkg865734533@163.com (X.W.)
2   Xi'an Center of Geological Survey, China Geological Survey, Xi'an 710054, China; henrylin@psu.edu (H.L.); xqiang@mail.cgs.gov.cn (Q.X.)
*   Correspondence: wangjiading029@163.com; Tel.: +86-135-7226-2307

**Abstract:** Landslide disasters have occurred frequently in the Chinese Loess Plateau (CLP) due to increased agricultural irrigation. To explore the spatiotemporal relationship between landslides and mechanisms at the Heifangtai terrace, the HFT irrigated area was selected as a typical case study to investigate the fundamental mechanism of the irrigation-induced landslide in the CLP. Multi-temporal remote sensing images, topographic maps, and unmanned aerial vehicle (UAV) photogrammetry data were used to investigate the evolution progress of landslides. Moreover, the evolution mechanism was discussed through topographic analysis, field monitoring, and laboratory testing. The results showed that erosion, collapse, and sliding had occurred at different scales and at different locations in the past 50 years. With an average retrogressive speed of $9.6 \times 10^3$ m$^2$ per year, the tableland decreased by $4.9 \times 10^5$ m$^2$ from 1967 to 2018, accounting for about 4.5% of its total area. Over 20 landslides and collapses were extracted in the Dangchuan section in the past four years. More than $5.48 \times 10^5$ m$^3$ of loess slipped with an average volume of 381 m$^3$ per day. The evolutionary process of the irrigation-induced landslide, which features retrogression, lateral extension, and clustering, began with local failure and ended in a series of slidings. The increase of groundwater level was a slow process, which is the reason for the lagged occurrence of the landslide. The influence of rainfall and irrigation on slope stability was greater than that of the periodic change of the groundwater level. The triggering effect of irrigation and rainfall on the landslide had a time lag due to slow loess infiltration, and the time response among irrigation, rainfall, and groundwater level was 4–6 months. Our findings provide guidance, concerning the planning and controlling of landslide disasters, which is of critical value for human and construction safety.

**Keywords:** agricultural irrigation; landslide-prone landscape; spatiotemporal relationship; sliding process; evolution mechanism



## 1. Introduction

Loess, an Aeolian, non-stratified deposit, is widely distributed around the world, but concentrated in the well-known Chinese Loess Plateau, with an arid to semi-arid climate. The behavior of loesses, especially those with high water sensitivities and structures, is intimately related to frequent loess landslides [1]. Moreover, the spatiotemporal distribution of loess landslides can be altered, to a certain extent, due to anthropogenic disturbances. For instance, in response to irrigation activities, about 200 landslides and 40 casualties have occurred in the Heifangtai (HFT) area since the 1970s [2–4]. What is worse is the high-density landslide-developing zone of landslides, about 10 km along the edge of the loess tableland, where subsequent landslides often follow previous landslides [5–9]. Such landslide-on-landslide effects are widespread in loess areas and often pose a serious threat

to large-scale casualties and property losses. Moreover, with the rapid development of urbanization and the economy in the Loess Plateau, human activities will contribute to more cases of the landslide-prone landscape, which leads to successive landslides.

Successive landslides can be causally related over the spatial and temporal distribution of existing landslides, as has been noted by recent studies. Such an assumption is supported by the fair agreement between natural experimental data and quantified analyses from theoretical models describing the overlap among landslides. These landslides are mostly distributed in the marginal regions of the loess tableland. The types and spatiotemporal distributions of HFT landslides in the Gansu province were particularly investigated [10–13], and the evolutionary process was discussed [9,14,15]. For example, Xu et al. [11,16] found that the scale of ground fissures reflected the evolution trend of the landslides, via their field investigations and geographic analysis of remote sensing data of the HFT area and the southern platform of the Jinghe River. Xue et al. [17] found that the average erosion rate of the back of the landslides were 4.47 m/y from 1977 to 1997, 3.46 m/y from 1997 to 2001, and 1.10 m/y from 2001 to 2010. The larger the annual irrigation volume was, the higher the erosion rate was, which approached a linear relationship. Liu et al. [10] generated multi-temporal landslide inventory maps and analyzed the landslide evolution features from December 2006 to November 2017 with C-, X-, and L-Band SAR datasets. More than 40 active loess landslides were mapped and their spatiotemporal evolution characteristics were revealed. Peng et al. [14] classified landslides in Heitai into five types and analyzed the distribution and failure mode of the landslides. They found that the retrogressive behavior resulted in a significant slope retreat along the edge of Heitai.

The region's best-studied representative successive landslides caused by human activities, in a site called HFT terrace, have been concerning. Significant advances have been made in the mechanisms of irrigation-induced landslides [18]. Qi et al. [9] believed that the retreat of the slope was caused by the retrogressive behavior of the loess flowslide, of which new failure developed at the main scarp of the preceding one. Peng et al. [8] analyzed the hydrological characteristics of Heitai and the effects of hydrological changes on the occurrence of different types of loess landslides, which was based on 43 boreholes and 51 2D electrical resistivity tomography (ERT) profiles. These studies summarized the types and spatiotemporal distribution of the irrigation-induced loess landslides and discovered the retrogressive behavior caused by the landslide. However, due to insufficient data before and after sliding, there is no unified understanding of the evolutionary process and related mechanism of these landslides.

In the existing research, the distribution of landslides are studied through data-driven susceptibility assessment approaches; the data sources are mainly data mining from high-resolution remote sensing images [9,19–22], UAV photogrammetry analysis [23–28], LIDAR data comparison [29], and synthetic aperture radar (SAR) interferometry analysis [10,30–34]. The evolution of landslides is based on geological analyses [35–39], long-term monitoring [40–42], remote sensing images [9], and other means. Extensive field investigation, satellite remote sensing, UAV aerial photography, and numeric analysis allowed researchers to identify the main kinematic features, the dynamic process, and the triggering mechanism of the landslide [43,44].

Not enough attention has been paid to the process mechanism and spatial and temporal distributions of these landslides. Systematic research on the associated spatial and temporal distribution of successive landslides in the HFT area is therefore of great significance for reducing economic losses and casualties in the Loess Plateau. Guided by earlier studies, the objective of this work was thus to give a more complete picture of the temporal and spatial features and associated mechanisms of the HFT landslide group. The relevance of the spatiotemporal distribution to landslide-prone landscapes using various methods are discussed in depth.

## 2. Materials and Methods

### 2.1. Study Area

HFT is located in Yanguoxia in northwest China, which covers an area of about 11.5 km², including Fengtai and Heitai, separated by Hulang Gully. Of which, the Fengtai tableland is in the west, with an area of 1.5 km², while the Heitai tableland is in the east, with an area of 9 km² (Figure 1a). From the perspective of regional tectonics, HFT is located in the eastern part of the middle Qilian and Lajishan-Wu fold belts of the Qilian orogenic belt. The lithological profile of the HFT is subdivided into four major stratigraphic units, from top to bottom, as follows (Figure 1b): upper Pleistocene Aeolian Malan loess (20–40 m), Alluvial clay (4–20 m), Fluvial gravels (3–7 m), and the mudstone of Cretaceous estuary group (more than 70 m).

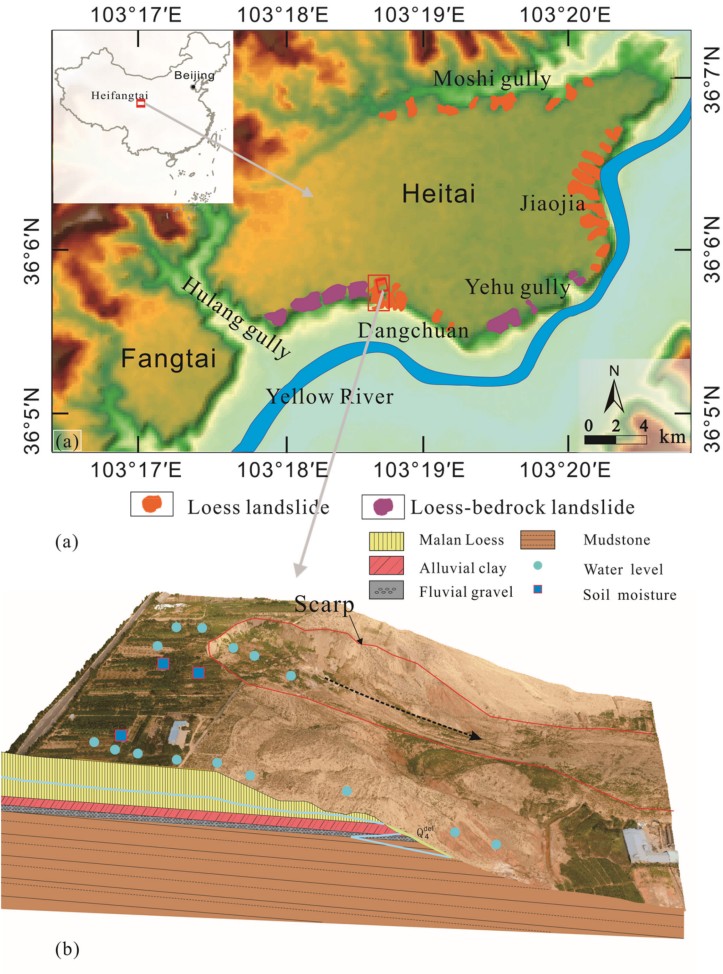

**Figure 1.** (**a**) Study area; (**b**) interpretative diagram of the lithological profile of the Dangchuan landslide cluster and site-monitoring layout.

In the early 1970s, the construction of a reservoir in the upper part of the Yellow River led to reconstruction issues for immigrants in the HFT area. The average annual evaporation (1689 mm) in HFT is about 6 times more than the average annual precipitation (277 mm). Hence, irrigation has become a dominant way to guarantee agriculture production, causing the groundwater level of the tableland to increase by more than 20 m at a rate of 0.18 m per year. Over the past 50 years, more than 200 landslides have occurred in HFT. Since 1980, the frequency of landslides has increased periodically within a 10-year cycle. On average, 3–5 landslides occur at HTF every year, which has led to immeasurable losses to the local people. Considering the movement types and material composition,

loess landslides distributed in Jiaojia, Dangchuan, Moshigou, with high-speed and long sliding distances, account for 69.5% of HFT landslides. Loess-bedrock landslides mainly occur mainly in the south margin of Heitai, which between Yehugou and Hulanggou, including "Huangci" and "Shuiguansuo" landslide clusters, with low speed and short sliding distances.

*2.2. Methods*

Five methods were used to characterize the spatiotemporal distribution of landslides: multi-temporal sensing images, topographic approach, and unmanned aerial vehicle (UAV) photogrammetry data to investigate the evolution progress of landslides. Moreover, field monitoring and laboratory tests were conducted to consider the associated mechanisms of the spatiotemporal distribution to landslide-prone landscapes.

2.2.1. Multi-Temporal Remote Sensing and Topographic Approach

A variety of medium and high-resolution remote sensing images were chosen for geomorphic change detection and quantification of topographic features presented in this paper (Table 1). Among them, the high-resolution images mostly come from the OrbView-3 satellite and Gaofen-1 satellite. Moreover, Google Earth images and deciphered KH-7, KH-4, and KH-9 remote sensing images were used. Due to the scarcity of high-resolution images from the 1980s to the early 2000s, some medium-resolution images (from Landsat and KATE) were also used.

**Table 1.** Remote sensing data adopted in the study.

| No. | Sources | Resolution | Period | Number of Images |
|---|---|---|---|---|
| 1 | KH-7(GAMBIT) | 0.6 m | March 1965–May 1967 | 4 |
| 2 | KH-4 (CORONA) | 2.7 m | November 1961–March 1971 | 12 |
| 3 | KH-9 (HEXAGON) | 6–9 m | November 1973–July 1975 | 2 |
| 4 | KATE 200 | 30 m | May 1982 | 1 |
| 5 | KFA-1000 | 3 m | August 1984 | 1 |
| 6 | Landsat-5 | 30 m | March 1987–March 2004 | 26 |
| 7 | Landsat-7 | 15 m (Band8) | March 2000–March 2005 | 10 |
| 8 | OrbView-3 | 1.0 m | February 2004–February 2007 | 6 |
| 9 | Gaofen-1 | 2.0 m | November 2013–February 2017 | 3 |
| 10 | Google images | 0.5–2 m | April 2002–October 2018 | 13 |

On the other hand, multi-temporal topographic maps were also used as they allowed researchers to interpret the topographic changes caused by landslides. As shown in Table 2, the first three types of data were from the Bureau of Surveying and Mapping of Gansu Province, the fourth from Xi'an Geological Survey Center, and the last from UAV photogrammetry.

**Table 2.** Topographic maps adopted in the study.

| No. | 1 | 2 | 3 | 4 | 5 |
|---|---|---|---|---|---|
| Map date | 1977 | 1997 | 2001 | 2010 | 2018 |
| Map scale | 1:10,000 | 1:10,000 | 1:10,000 | 1:1000 | 1:500 |

2.2.2. UAV Photogrammetry

UAV photogrammetry was used to obtain high-resolution orthophoto images and digital elevation data from different periods. Considering the frequency of landslides and the availability of human power, 2–3 missions were conducted every year from 2015 to 2019. In 2015–2017, the DJI Phantom 3 and DJI Phantom 4 were used in photogrammetry, and the DJI Phantom 4 RTK was adopted in 2018. The maximum image resolution of the DJI Phantom 3 camera was 4000 × 3000, while Phantom 4 was 5472 × 3648. A total of

13 missions were completed until 2019 (Table 3), including 10 large-scale mappings and 3 small-scale mappings. Of which, large areas referred to the entire landslide tableland, small-scale areas referred to the Dangchuan landslide and the Jiaojia landslide. The ground resolution of 6.0 cm/pixel was widely used in flights while 1.0–2.0 cm was commonly employed for mappings to compare crack changes. The digital orthophoto map (DOM) image and a digital surface model (DSM) were generated by way of the pre-set control points. More than 200 control points were set using real time kinematic (RTK), with an absolute accuracy of 2–4 m. The ground data of the study area were obtained by means of UAV accompanied by ground control points. Subsequent data processing was performed using Agisoft Photo-Scan. To assess the accuracy of DSMs and orthophotos, approximately 30% of the GCPs were used as checkpoints. The root-mean-square error (RMSE) values were about 2–15 cm in the horizontal direction (XY) and 2–20 cm in the vertical direction (Z).

**Table 3.** Overview of all performed UAV missions for the study area.

| UAV Mission | 1st | 2nd | 3st | 4th | 5th | 6th | 7th | 8th | 9th | 10th | 11th | 12th | 13th |
|---|---|---|---|---|---|---|---|---|---|---|---|---|---|
| Flight date | May 2015 | July 2015 | June 2016 | November 2016 | February 2017 | August 2017 | October 2017 | April 2018 | August 2018 | November 2018 | January 2019 | May 2019 | July 2019 |
| Number of flights | 8 | 12 | 39 | 28 | 36 | 15 | 55 | 62 | 41 | 18 | 22 | 19 | 20 |
| Area covered ($km^2$) | 0.98 | 2.25 | 7.37 | 6.71 | 6.99 | 2.11 | 14.12 | 16.04 | 10.93 | 8.06 | 8.12 | 8.08 | 8.10 |

### 2.2.3. Site Evaluation and Laboratory Analysis

The distribution and variation of the seepage field near the slope are the main factors affecting the development of landslides [45–47]. Hence, fifteen pore pressure sensors installed in Dangchuan landslide were employed to monitor the groundwater at different locations. Moreover, soil moisture sensors were installed in farmlands with different depths (0.1, 0.2, 0.5, 1.0, 2.0, 3.0, and 5 m) to assess the effect of irrigation on soil moisture (Figure 1b).

The undisturbed soil blocks used in this paper were collected from a three-meter deep test well. The samples needed to be pre-trimmed (39.1 mm in diameter and 80 mm in height) prior to laboratory testing. The samples were tested for triaxial shear strength at different moisture contents. In addition, uniaxial tensile tests with different water contents were performed. Finally, the occurrence mechanism was discussed and analyzed in combination with numerical simulations.

## 3. Results

### 3.1. Topographic Change in Landslide Area

Figure 2a shows the results of the degradation distance performed on tableland edge from 1967 to 2018. Numerous landslides occurred on the northeast, southeast, and southwest edges of the Heitai tableland, which finally developed into three concentrated landslide clusters (Dangchuan, Jiaojia, and Moshi) (Figure 2a). The change in tableland edge from 1967 to 2018 was associated with irrigation-induced loess landslides. The three typical tableland boundary sections marked by blue lines are shown in Figure 2b.

Figure 2b compares the images in zone II at different periods. From 2002 to 2018, about 29 landslides occurred, with a total length of 190 m. It is worth noting that three landslides (width: ~100 m, length: ~80) were found in Figure 2b2. Moreover, four large landslides and five small landslides were identified in the same area in 2018 (Figure 2b3).

The degradation characteristics of the tableland edge were extracted from 1967 to 2018 (Figure 3). A considerable reduction of land ($4.9 \times 10^5$ $m^2$) in Heitai is detected, representing about 4.5% of the total. In particular, the reduced area in the three zones is $2.66 \times 10^5$ $m^2$, $4.97 \times 10^4$ $m^2$, and $2.79 \times 10^4$ $m^2$, respectively, noting that land area gradually decreased after 1982, primarily due to irrigation activities. More specifically, the land regressive speed of Heitai fluctuates in a sinusoidal pattern, peaking in 1991–1993

and 2012–2016, respectively. Zone I also has a similar fluctuation pattern. Moreover, the landslide clusters of zone I was active from 1982 to 2016, and gradually disappeared after 2016. In contrast, both zone II and III become high landslide-prone landscapes after 2010.

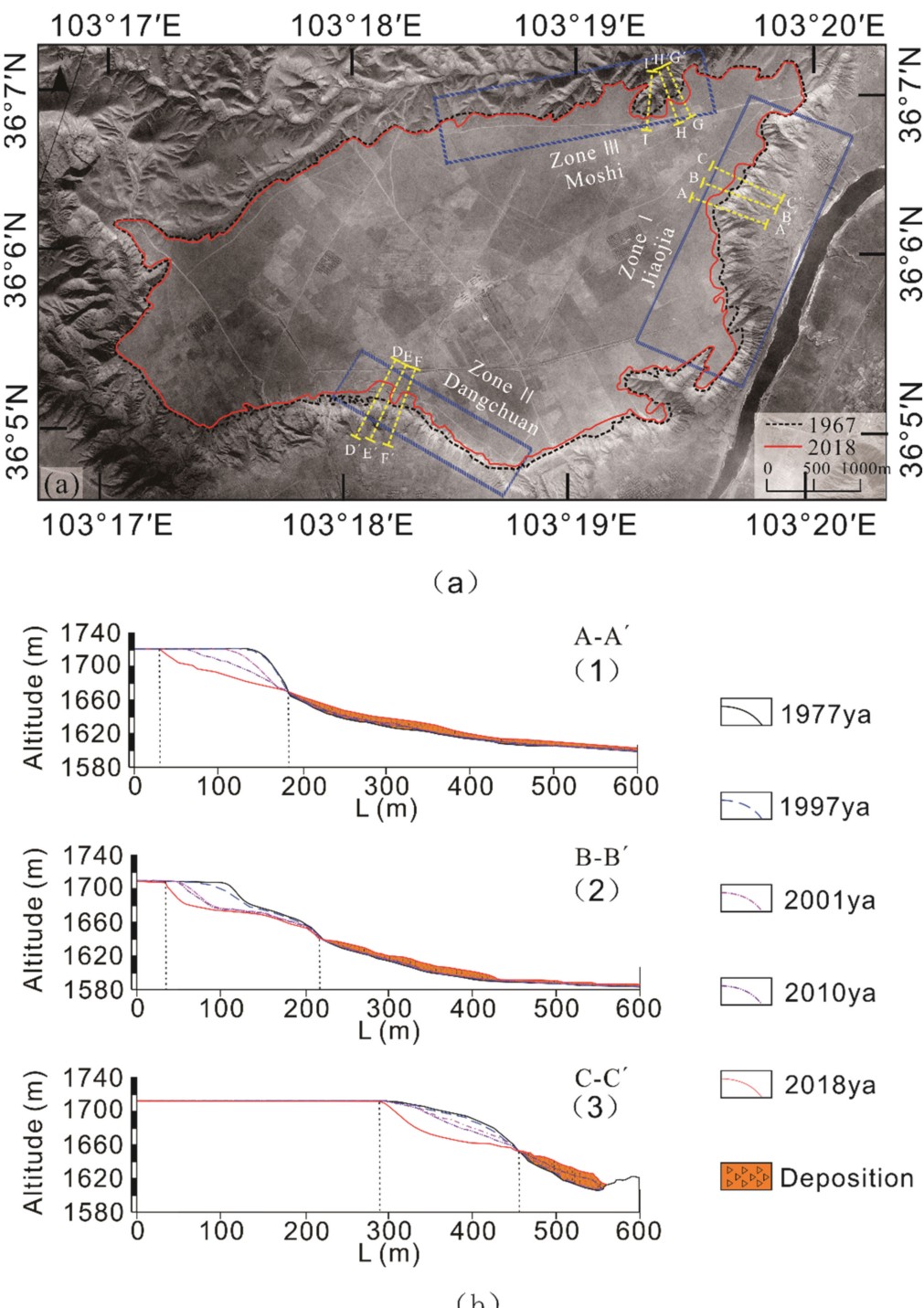

**Figure 2.** Evolution of tableland edge from 1967 to 2018. (**a**) The results of degradation distance performed; (**b**) the images in zone II at different periods.

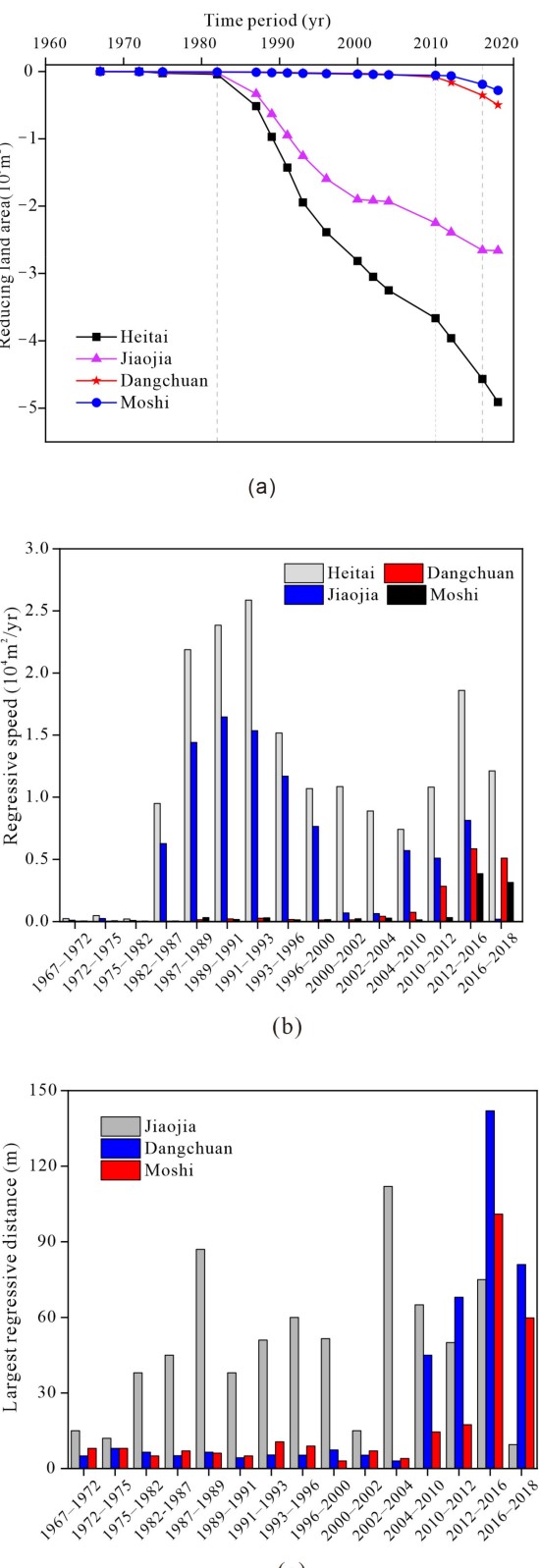

**Figure 3.** Evolution characteristics of tableland edge. (**a**) Degradation area; (**b**) degradation speed; (**c**) maximum degradation distance.

### 3.2. Evolutionary Process of Landslide

Nearly all successive landslides occurred in the Dangchuan area in recent years; thus, the current study in this article mainly focused on the Dangchuan landslide cluster. Over

the past five years, more than twenty landslides have been recorded along a one-kilometer-long edge of the tableland. The landslide morphology in Dangchuan from 2015 to 2019 is shown in Figure 4.

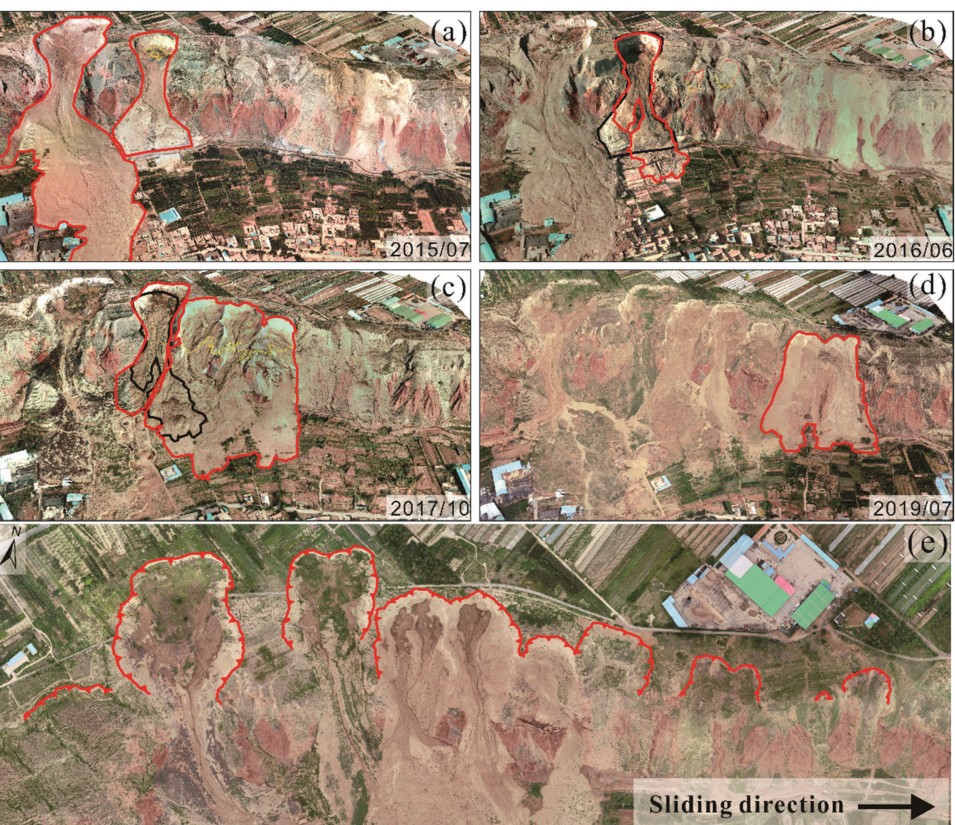

**Figure 4.** Reconstructions of Dangchuan from 2015 to 2019: (**a–e**) three-dimensional reconstructions. Slidings extracted from a digital surface model (DSM) and orthophoto image.

The evolutionary process of landslides induced by irrigation began with local failure and developed to a single sliding. A large number of single slidings were superimposed to form a group of landslides, which caused the tableland surface to retreat. Based on data analysis reported above and field observations, we identified the following five stages of landslide evolution in the study area:

(1)    Basic geological environment of the slope. The deep water table in the study area before irrigation, the high water sensitivity of the soil, and the distribution of initial gullies create conditions for avalanches, etc. Concentrated rainstorms during the year increased lateral erosion of the river [48], and concentrated rainstorms promoted erosion.

(2)    Provision of irrigation breeding factors. With the construction of many irrigation projects, long-term flood irrigation increased groundwater recharge and formed a new cycle of groundwater recharge, runoff, and discharge. The groundwater level rose sharply, and the descending spring appeared at the edge of the tableland. A large number of collapsible cracks appeared on the tableland, which accelerated irrigation water infiltration and loess suffusion. Increased debris also aggravated the development of gullies.

(3)    The occurrence of landslides. Irrigation changed the water table and the hydraulic properties of soil, which led to the continuous collapse around the slope, the decrease of stability, and the increase of the number of springs in front of the landslide.

(4)    The number of landslides and the increase of sliding displacement. The occurrence of the initial landslide removed the main slide direction and the surrounding soil.

At the same time, the number of springs in the lower part increased constantly, and the spring took away many loose deposits from the slide bed, which increased the displacement of the slide direction. The trailing edge of landslides constantly receded and engulfed roads, villages, and farmland because of the movement of slide bodies.

(5)　Formation of great erosion gully. Under the action of irrigation, landslides continuously occurred and expanded; several landslides were merged, the original platform receded continuously, the sliding direction of several landslides and the connection of surrounding grooves were formed, and a new large-scale erosion ditch was formed.

It is observed that the evolution of irrigation-induced landslides had the characteristics of retrogression, lateral expansion, and clustering. The back of landslides kept retrogressing, showing a progressive destruction feature. Such an evolution of landslide occurrence in the CLP led to the evolutionary change of the CLP land surface.

## 4. Discussion

The occurrence of landslides is affected by various factors, but the occurrence of landslides in this area is mainly affected by irrigation, followed by the effect of rainfall [49–51]. The average annual precipitation of HFT was only 277 mm. In contrast, the annual irrigation water volume was about $5.9 \times 106$ m$^3$, adding an extra water supply of 513 mm/m$^2$, nearly 1.8 times as much as the annual rainfall. Flood irrigation made the infiltration volume increased sharply, causing the groundwater level to rise significantly. When the water level in the center of the tableland rose to about 15 m, the landslide began. That was about 20 years later than the start of irrigation. By the end of 2018, the groundwater level at the edge of the tableland had risen by 15.8 m, respectively (Figure 5). While the landslide had occurred more than 100 times, the tableland area had decreased by $4.9 \times 10^5$ m$^2$. Therefore, these landslides were caused by the long-term effects of the rising groundwater level. Irrigation led to an increase in soil moisture. According to Figure 6a, the soil moisture within 0.5–3 m increased slowly. Rainfall could also increase the soil moisture, especially continuous rainfall. The rainfall and irrigation had a lag effect on soil moisture, and the time lag increased with the increase of depth. As a result, the landslide always lagged behind the rainfall and irrigation. The time response among irrigation, rainfall, and groundwater level was 4–6 months, so the groundwater level was the highest from February to April (Figure 6b). The rise of the groundwater level also led to the increase of deep soil moisture, which was confirmed by long-term ERT monitoring in Heitai [8,42]. The monitoring results show that the groundwater rising speed of the Dangchuan Section is 0.30 m per year.

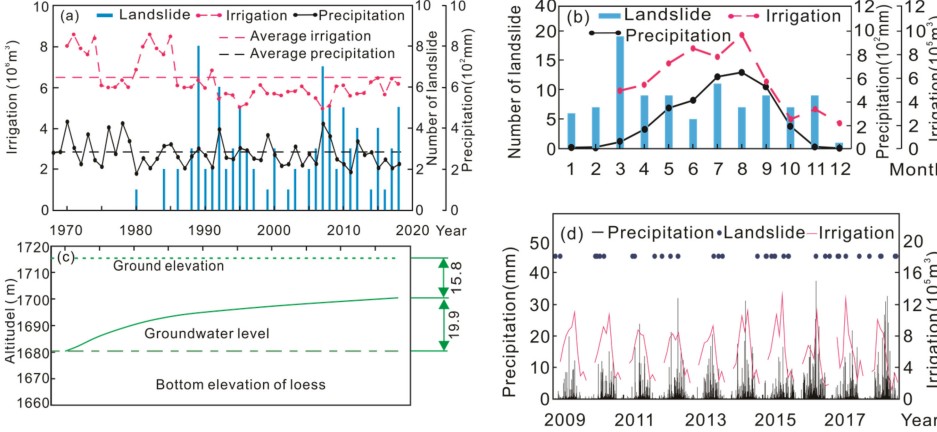

**Figure 5.** Statistical charts of precipitation, irrigation, groundwater level, and landslide; (**a**,**c**) charts of interannual variety from 1968 to 2018; (**b**) chart of inter-monthly variation; (**d**) diagram chart of precipitation, irrigation, and landslide from 2009 to 2018.

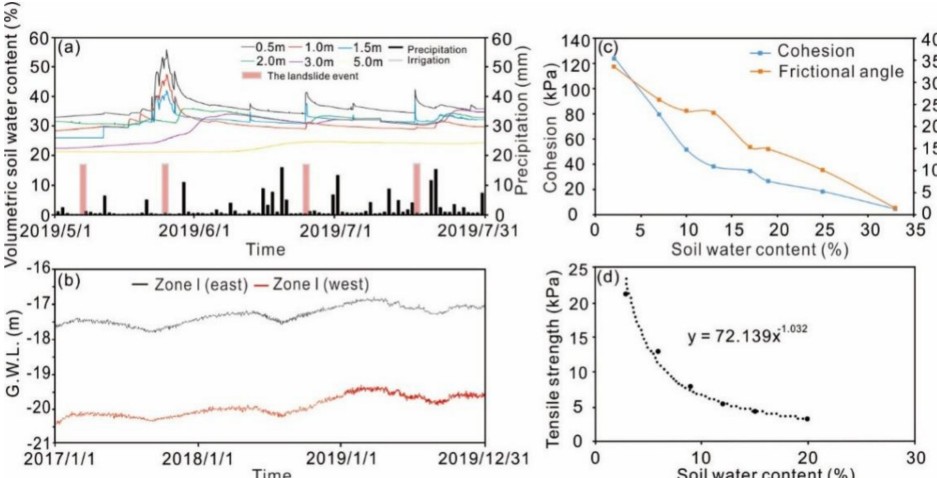

**Figure 6.** Irrigation-effect on groundwater level, soil moisture and soil strength: (**a**) changes of soil moisture after irrigation and precipitation; (**b**) GWL, Groundwater level change curve with time; (**c**) shear strength of triaxial shear test; (**d**) tensile strength of uniaxial tensile test).

The rainfall and irrigation could also change the physical and mechanical properties of loess. According to our tests, the shear strength of samples with 6% soil moisture was twice as much as those with 30% soil moisture (Figure 6c). With the increase of soil moisture, the tensile strength decreased exponentially (Figure 6d). The tensile strength of the loess is about 15% of its cohesion. The results of unsaturated soil tests show that the cohesion increased linearly with the increase of matric suction, but the effective internal friction angle was constant [51,52]. This shows that irrigation could reduce loess strength by increasing the soil moisture of slope. The peak value and residual shear strength of loess can be significantly reduced by lithification with irrigation water [53,54]. Therefore, irrigation water played a crucial role in reducing the shear strength of the slide surface.

Unsaturated seepage stability analysis was used to analyze the effect of different factors on the formation of the landslide. The methods of unsteady flow and unsaturated stability were adopted in the numerical analysis via using GeoStudio, a professional geotechnical software. The material parameters and analysis process were the same as in our previous papers [45,55,56]. Two cross-sections in Dangchuan and one cross-section in Jiaojia were selected to establish the models (Figure 7a,b). The influence of the rise of the groundwater level on the stability was simulated by setting a time-varying groundwater level boundary on the left side of the models. The analysis results of model 2 (Figure 7c) show that the stability was 1.51 in 1967 and decreased to 0.98 in 2018. The initial stability of model 1 was 0.14 lower than that of model 2. However, it surpassed the latter by 0.05 after 51 years later. The stability of the concave slope decreased faster than that of a convex slope, owing to different geometry and seepage field of the slope. Previous studies [55] have also shown that slope stability decreased by 0.1 as groundwater level rose by 5 m. The influence of rainfall on slope stability was simulated by setting rainfall boundary (Figure 7d,f) in the upper part of model 1. Figure 7f shows the change of slope stability during the rainy season (June–August). The results show that rainfall could reduce the stability of the slope slightly, and it had a lag effect. Figure 7f shows that, with the arrival of the rainy season, the slope stability decreased, reaching the lowest level in late August. Moreover, it slightly increased as the rainy season came to its end. Although rainfall had a small impact on the slope safety factor, most slopes had become unstable, as the groundwater level had risen by more than 20 m. The landslide could be triggered by a rainstorm or persistent rain. Due to the lack of irrigation records at the slope where the model was located, we simplified the irrigation to once a month from March to December (Figure 7e). The calculation results show that flood irrigation could cause slope instability (Figure 7e). The effect of flood irrigation on slope stability was higher than that of rainfall. The influence of rainfall and

irrigation on slope stability was greater than that of the periodic change of groundwater level (Figure 7f,e). The periodic change of the groundwater level is mainly caused by rainfall and irrigation, which has a long lag process. Gu et al. [45] analyzed the effect of different irrigation locations on slope stability. The results show that when the distance between flood irrigation and the edge of the tableland was less than 60 m, the influence of irrigation on slope stability became greater, and the stability of the slope decreased, almost in a straight line. When the distance was less than 25 m, the stability of the slope decreased from 1.10 to 0.98 within three years. It reveals that the closer the irrigation location was to the edge of the tableland, the more likely the irrigation would affect the soil near the potential sliding surface. The above analysis demonstrated that irrigation or rainfall had a triggering effect on the landslide and had a long time lag. The decline of shear strength of loess caused by loess desalination could also reduce slope stability. However, leaching is a slow process (desalination rate < 50% in more than 40 years), which is one of the reasons for the lagged occurrence of the loess landslides [57].

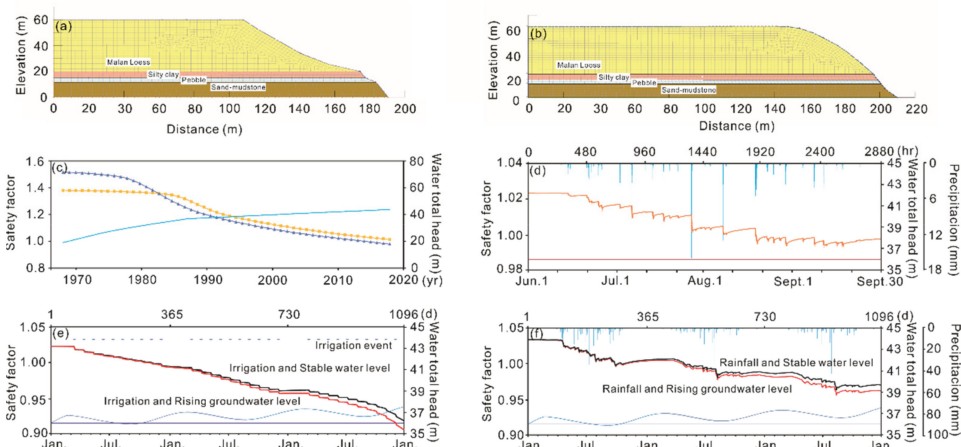

**Figure 7.** Results of saturated-unsaturated seepage and stability analysis: (**a**,**b**) numerical analysis model; (**c**). Variation in safety factor due to groundwater level change; (**d**,**f**) variation in safety factor due to precipitation (H1, stable water table; H2, rising water table; Fs, safety factor); (**e**) variation in safety factor due to irrigation.

From the above analysis, we can see that the landslide has continuously developed and changed. Moreover, the side erosion of gully and collapse has been happening at different scales and at different locations. As the loess collapsed and disintegrated when mixed with water [46], collapsible cracks and sinkholes often appeared in the loess tableland with irrigation, especially at the edge of the irrigation area (Figure 8) [58–61]. Moreover, the loess was of low tensile strength [47,51], which is another reason for the occurrence of dense cracks on the edge of the tableland [62]. As time went on, cracks kept expanding and small collapses occurred continuously on the sides and back of the gully. Moreover, small-scale slidings occurred on the back and sides of the landslides. Loose deposits kept moving downward under the action of gravity and flowing water. In this process, there was also a small amount of large-scale sliding. Such sliding generally occurred at the back of the existing landslide. Landslides affected the topography of the middle part of the slope greatly. On the one hand, some debris accumulated in the channel. On the other hand, the debris accelerated continuously during the sliding process, resulting in the erosion of the gully. Therefore, some silty clay layers and sandy mudstone layers slipped together with the upper debris, and a large amount of debris was transferred to the lower part of the landslide. The debris dropped by the landslide and flew downwards along the slope, accumulating at the toe of the slope. The extent of the deposit was related to the scale of the landslide. Some of the debris dropped by landslides rushed into the Yellow River and made huge waves [12] because of the lack of river terraces. Therefore, the occurrence of this landslide complex is the combined effect of various factors (soil, groundwater, and rainfall).

In addition, natural disasters are physical phenomena that are active during geological periods, and it is extremely difficult to prevent them from occurring. Although the study area is in a rural area, landslides continue to occur, and the slopes where they do not occur are still inhabited by large numbers of people and planted with fruit trees and crops. In the long term, the relevant authorities should link the occurrence of landslides and land planning closely, relocate people where there is already an obvious hazard phenomenon, and who are not in a safe area, and stop all planting activities to avoid casualties and unnecessary property damage.

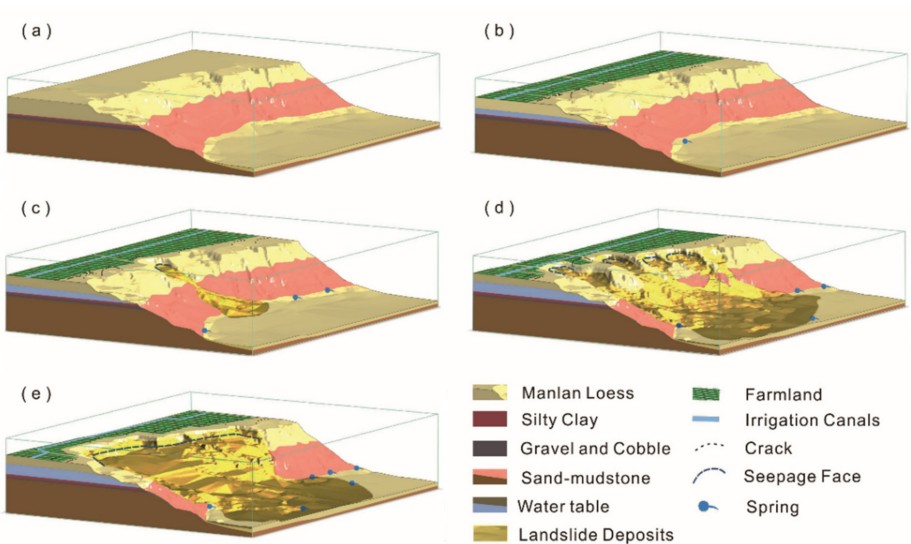

**Figure 8.** Morphological evolutionary processes of irrigation-induced landslide: (**a**) initial stage; (**b**) irrigation spread and the water table rose sharply; (**c**) the occurrence of landslides; (**d**) retrogression and lateral expansion of landslides; (**e**) landslide clustering and landscape change.

## 5. Conclusions

The main findings of this study can be summarized as follows:

(1) According to historical images, that painful period of the Dangchuan section started after 2010. Landslides first occurred at the end of gullies, and then sliding occurred again on the back or side of existing landslides. The nearer the sliding was to the center of the tableland, the higher the occurrence of landslides and the larger the scale of landslides. With an average retrogressive speed of $9.6 \times 10^3$ m$^2$ per year, the tableland decreased by $4.9 \times 10^5$ m$^2$ from 1967 to 2018, accounting for about 4.5% of its total area. The reduced land of the zones in Jiaojia, Dangchuan, and Moshi was $2.66 \times 10^5$ m$^2$, $4.97 \times 10^4$ m$^2$, and $2.79 \times 10^4$ m$^2$, respectively.

(2) UAV mapping results of the Dangchuan section showed that there had been nearly 20 slidings at the edge of the tableland from July 2016 to July 2019. There were more than $5.48 \times 10^5$ m$^3$ of slipped loess, with an average volume of 371 m$^3$ per day. A series of cracks extended at the back edge of the landslide, and multiple small collapses occurred at the side and back of the gully. Some small-scale slidings and a few large-scale slidings also occurred at the back and side of the landslide. The debris dropped by the landslides, flowed downwards along the slope, and eventually accumulated at the front edge of the slope. The scale of the landslide determined the range of accumulation.

(3) The closer the irrigation position was to the edge of the tableland, the easier the irrigation water would affect the soil near the potential sliding surface, eventually resulting in slope instability. The groundwater level at the edge of the slope in Jiaojia (east) was about 5–6 m shallower than that in Dangchuan (south), and about 1–2 m shallower than that in Moshi (north). As the groundwater level continued to rise,

the landslides developed from east to north and south in Heitai. The increase of groundwater level was a slow process, which is the reason for the lagged occurrence of the landslide.

(4)    The evolution of irrigation-induced landslides led to the change in geomorphology. The evolutionary process began with local failure, and then developed to a single sliding. Substantial slidings merged to form landslide groups, which caused the tableland surface to retreat. The back of the landslides kept retrogressing, which showed a progressive destruction feature. The main causes of irrigation-induced landslides were the rise of groundwater level and the decrease of loess intensity. The decrease of loess strength lay in the humidification and desalination of loess. Moreover, the influence of rainfall and irrigation on slope stability was greater than that of the periodic change of the groundwater level. The triggering effect of irrigation and rainfall on the landslide had a time lag due to slow loess infiltration, and the time lag among irrigation, rainfall, and groundwater level was 4–6 months.

**Author Contributions:** T.G.: analyses, writing; J.W.: data; H.L.: analyses of the field geological phenomena; Q.X.: writing—original draft preparation; B.S.: indoor tests; J.K.: drone aerial photography; J.S.: drone aerial photography; C.W.: drone aerial photography; F.Z.: drone aerial photography; X.W.: drone aerial photography. All authors have read and agreed to the published version of the manuscript.

**Funding:** This research was funded by the National Natural Science Foundation of China (grant numbers, 41772285, 41630639, 41530640), National Key R&D Program of China (2018YFC1504703), and the State Key Laboratory of Continental Dynamics.

**Institutional Review Board Statement:** Not applicable.

**Informed Consent Statement:** Informed consent was obtained from all subjects involved in the study and written informed consent has been obtained from the patients to publish this paper.

**Data Availability Statement:** According to the joint decision of all authors, the data cannot be used to other article.

**Conflicts of Interest:** The authors declare no conflict of interest.

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
