# Peer review of "The Spatiotemporal Relationship between Landslides and Mechanisms at the Heifangtai Terrace, Northwest China"

_water, doi:10.3390/w13223275_

Round 1

Reviewer 1 Report

The authors write about relationships and mechanisms in terms of time, while the results they present are not supported by specific results. The paper lacks a multidimensional data analysis that would clearly confirm the obtained calculation results, which should be supplemented. 

Author Response

Manuscript ID: water-1359995

Manuscript Title:   On the spatiotemporal relationship between landslides and mechanisms on the Heifangtai terrace, Northwest China

Dear Editor,

We would like to thank you for sending our work for review, and the reviewers for their insightful and constructive comments. We have revised our manuscript, as highlighted in red color in the main body according to comments of the reviewers. We hope that this revised version will satisfy the reviewers.

The point by point responds to the reviewer’s comments are as follows:

[Question 1]

The authors write about relationships and mechanisms in terms of time, while the results they present are not supported by specific results. The paper lacks a multidimensional data analysis that would clearly confirm the obtained calculation results, which should be supplemented.

[Answer 1]:

Thanks for your good and positive comments. Your suggestion should be adopted. This paper investigates the mechanism of landslide occurrence and its evolution from a temporal perspective, using data from the temporal perspective of landslide clusters, irrigation and rainfall data over many years, data from indoor experiments, and validation of numerical analysis to develop the paper's research. In addition, more data are still being collected.

Except for the reviewers’ comments, we also made some changes in the revised manuscript. These changes will not influence the content and framework of the paper. And here we did not list the changes but marked in red in revised paper.

We appreciate for Editors/Reviewer’s warm work earnestly, and hope that the corrections will meet with approval. Please feel free to contact us with any questions and we are looking forward to your consideration.

Once again, thank you very much for your comments and suggestions.

Best regards,

Tianfeng Gu, Jiading Wang, Herry Lin, Qiang Xue, Bin Sun, Jiaxu Kong, Jiaxing Sun, Chenxing Wang, Fanchen Zhang.  

Reviewer 2 Report

The manuscript entitled “On the spatiotemporal relationship between landslides and mechanisms on the Heifangtai terrace, Northwest China”, by T. Gu, J. Wang, H. Lin, Q. Xue, B. Sun, J. Kong, J. Sun, C. Wang, F. Zhang and X. Wang, presents an interesting work.

In general, the manuscript should be acceptable for publication but some serious problems must be repaired prior to publication. It needs some significant improvement. Some suggestions are as follows:

  1. Please follow the journal author instructions. It would be useful for the reader to follow the classical text structure (i.e. Introduction-methodology-results-discussion-conclusions). A better presentation of your results and an extensive discussion would improve your paper.
  2. I suggest to separate the section “3. Results and Discussion”.
  3. It would be useful to be described the aim of this paper.
  4. The English language usage should be checked by a fluent English speaker. It is suggested to the authors to take the assistance of someone with English as mother tongue.
  5. In all maps you must put coordinates.
  6. Please be careful with the spaces between the words.
  7. Correct references in the text and the reference list according to the journal’s format. Please format the references’ list by using the correct journal abbreviations. See the following link: https://images.webofknowledge.com/images/help/WOS/A_abrvjt.html
  8. You could enrich the scientific literature.
  9. Please justify convincingly why this manuscript (method, thematology etc) connected with water’s content and scope. Perhaps the using of proper literature would be helpful. Eg: Troncone, A.; Pugliese, L.; Parise, A.; Conte, E. Prediction of Slow-Moving Landslide Mobility Due to Rainfall Using a Two-Wedges Model. Water 2021, 13, 2030. https://doi.org/10.3390/w13152030
  10. The authors could make discussion about the relationship between landslide assessment and planning. See the following publications: Skilodimou, H.D.; Bathrellos, G.D. Natural and Technological Hazards in Urban Areas: Assessment, Planning and Solutions. Sustainability 2021, 13, 8301. https://doi.org/10.3390/su13158301.

Author Response

Manuscript ID: water-1359995

Manuscript Title:   On the spatiotemporal relationship between landslides and mechanisms on the Heifangtai terrace, Northwest China

Dear Editor,

We would like to thank you for sending our work for review, and the reviewers for their insightful and constructive comments. We have revised our manuscript, as highlighted in red color in the main body according to comments of the reviewers. We hope that this revised version will satisfy the reviewers.

The point by point responds to the reviewer’s comments are as follows:

[Question 1]

Please follow the journal author instructions. It would be useful for the reader to follow the classical text structure (i.e. Introduction-methodology-results-discussion-conclusions). A better presentation of your results and an extensive discussion would improve your paper.

[Answer 1]:

Thanks for your good and positive comments. Your suggestion should be adopted. We have restructured the full text.

[Question 2]

I suggest to separate the section “3. Results and Discussion”.

[Answer 2]:

Thanks for your good and positive comments. Your suggestion should be adopted. The third chapter in the article has been separated from results (in line 187) as well as discussion (in line 269). Furthermore, we have added the discussion in line 270-271.

[Question 3]

It would be useful to be described the aim of this paper.

[Answer 3]:

Thanks for your good and positive comments. Your suggestion should be adopted. In the article abstract, we have added the research objectives in line 13-14.

[Question 4]

The English language usage should be checked by a fluent English speaker. It is suggested to the authors to take the assistance of someone with English as mother tongue.

[Answer 4]:

In order to improve the text, we have sought revisions from native English speakers.

[Question 5]

In all maps you must put coordinates.

[Answer 5]:

Thanks for your good and positive comments. Based on your valuable comments, we have added location coordinates to all maps.

[Question 6]

Please be careful with the spaces between the words.

[Answer 6]:

Thanks for your good and positive comments. Your suggestion should be adopted. Spaces between words have been removed.

[Question 7]

Correct references in the text and the reference list according to the journal’s format. Please format the references’ list by using the correct journal abbreviations. See the following link: https://images.webofknowledge.com/images/help/WOS/A_abrvjt.html.

[Answer 7]:

Thank you very much for your valuable suggestion, we have used journal abbreviations for the full text.

[Question 8]

You could enrich the scientific literature.

[Answer 8]:

Thank you for your good and positive comments. Your suggestions should be adopted. in order to meet the needs of the article,We have enriched the text with these references :

  1. Bathrellos, G.D.; Skilodimou, H.D. Land use planning for Natural Hazards. Land. 2019, 8, 128-131. https://doi.org/10.3390/land8090128
  2. Trieu,; Bergström, P.; Sjödahl, M.; Hellström, J. G. I.; Andreasson, P.; Lycksam, H. Photogrammetry for Free Surface Flow Velocity Measurement: From Laboratory to Field Measurements. Water. 2021,13,1675-1688. https://doi.org/10.3390/w13121675.
  3. He, Y,;Sun,R,;Xu, Z,; Tang,  The Dynamic Change and Effect of Rainfall Induced Groundwater Flow. Water. 2021, 13, 2625-2638. https://doi.org/10.3390/w13192625.
  4. Wu, C.; Lin, C. Spatiotemporal Hotspots and Decadal Evolution of Extreme Rainfall-Induced Landslides: Case Studies in Southern Taiwan. Water.2021, 13, 2090-2108. https://doi.org/10.3390/w13152090
  5. Xu, L.;Yan, D.D. The groundwater responses to loess flowslides in the Heifangtai platform.  Eng. Geol. Environ. 2019, 78(7), 4931-4944. https://doi.org/10.1007/s10064-018-01436-4.
  6. Qi, X.; Xu, Q.; Liu, F.Z. Analysis of retrogressive loess flowslides in Heifangtai, China. Eng. Geol. 2018, 236, 119-128. https://doi.org/10.1016/j.enggeo.2017.08.028.
  7. Hou, X.K.; Vanapalli, S.K.; Li, T.L. Water infiltration characteristics in loess associated with irrigation activities and its influence on the slope stability in Heifangtai loess highland, China. Eng. Geol. 2018, 234, 27-37. https://doi.org/10.1016/j.enggeo.2017.12.020.

[Question 9]

Please justify convincingly why this manuscript (method, thematology etc) connected with water’s content and scope. Perhaps the using of proper literature would be helpful. Eg: Troncone, A.; Pugliese, L.; Parise, A.; Conte, E. Prediction of Slow-Moving Landslide Mobility Due to Rainfall Using a Two-Wedges Model. Water 2021, 13, 2030. https://doi.org/10.3390/w13152030

[Answer 9]:

Thank you very much for your valuable comments on the article, which should be taken into account. According to our past research on the landslide and the findings of researchers with related expertise in the same field, a large number of landslides in the area occur as a result of irrigation action and a small number of landslides are induced by rainfall when there is no irrigation action, so the manuscript (methodology, thematics, etc.) is relevant to the content and scope of water. At the same time, we have added relevant references in the text.

[Question 10]

The authors could make discussion about the relationship between landslide assessment and planning. See the following publications: Skilodimou, H.D.; Bathrellos, G.D. Natural and Technological Hazards in Urban Areas: Assessment, Planning and Solutions. Sustainability 2021, 13, 8301. https://doi.org/10.3390/su13158301.

[Answer 10]:

Thank you for your good and positive comments. Your suggestions should be adopted. This has been discussed in terms of landslide assessment and planning, as can be seen in the discussion section of the article where in line 378-386.

Except for the reviewers’ comments, we also made some changes in the revised manuscript. These changes will not influence the content and framework of the paper. And here we did not list the changes but marked in red in revised paper.

We appreciate for Editors/Reviewer’s warm work earnestly, and hope that the corrections will meet with approval. Please feel free to contact us with any questions and we are looking forward to your consideration.

Once again, thank you very much for your comments and suggestions.

Best regards,

Tianfeng Gu, Jiading Wang, Herry Lin, Qiang Xue, Bin Sun, Jiaxu Kong, Jiaxing Sun, Chenxing Wang, Fanchen Zhang.  

Reviewer 3 Report

The manuscript entitled “On the spatiotemporal relationship between landslides and mechanisms on the Heifangtai terrace, Northwest China presents an interesting work.

The manuscript could be acceptable for publication, but it needs improvement. I suggest:

  1. It is suggested to the authors to take the assistance of someone with English as mother tongue.
  2. You could disconnect the chapter 3.
  3. It would be useful to be described the aim of this paper.
  4. Please put coordinates in all maps.
  5. Be careful with the spaces between the words.
  6. Use the journal abbreviations.
  7. You could use recent useful references.
  8. You could use the related literature from this journal, like:

- Wu, C.; Lin, C. Spatiotemporal Hotspots and Decadal Evolution of Extreme Rainfall-Induced Landslides: Case Studies in Southern Taiwan. Water 2021, 13, 2090. https://doi.org/10.3390/w13152090

- Bathrellos, G.D.; Skilodimou, H.D. Land Use Planning for Natural Hazards. Land 2019, 8, 128. https://doi.org/10.3390/land8090128

Author Response

Manuscript ID: water-1359995

Manuscript Title:   On the spatiotemporal relationship between landslides and mechanisms on the Heifangtai terrace, Northwest China

Dear Editor,

We would like to thank you for sending our work for review, and the reviewers for their insightful and constructive comments. We have revised our manuscript, as highlighted in red color in the main body according to comments of the reviewers. We hope that this revised version will satisfy the reviewers.

The point by point responds to the reviewer’s comments are as follows:

[Question 1]

It is suggested to the authors to take the assistance of someone with English as mother tongue.

[Answer 1]:

Thanks for your good and positive comments. Your suggestion should be adopted. In order to improve the text, we have sought revisions from native English speakers.

[Question 2]

You could disconnect the chapter 3.

[Answer 2]:

Thanks for your good and positive comments. Your suggestion should be adopted. The third chapter in the article has been separated from results (in line 187) as well as discussion (in line 269). Furthermore, we have added the discussion in line 270-271.

[Question 3]

It would be useful to be described the aim of this paper. 

[Answer 3]:

Thanks for your good and positive comments. Your suggestion should be adopted. In the article abstract, we have added the research objectives in line 13-14.

[Question 4]

Please put coordinates in all maps. 

[Answer 4]:

Thanks for your good and positive comments. Based on your valuable comments, we have added location coordinates to all maps.

[Question 5]

Be careful with the spaces between the words. 

[Answer 5]:

Thanks for your good and positive comments. Your suggestion should be adopted. Spaces between words have been removed.

[Question 6]

Use the journal abbreviations.

[Answer 6]:

Thank you very much for your valuable suggestion, we have used journal abbreviations for the full text.

[Question 7]

You could use recent useful references.

[Answer 7]:

Thanks for your good and positive comments. Your suggestion should be adopted. In order to meet the needs of the article, we have used the most recent references in the text.

[Question 8]

You could use the related literature from this journal, like:

- Wu, C.; Lin, C. Spatiotemporal Hotspots and Decadal Evolution of Extreme Rainfall-Induced Landslides: Case Studies in Southern Taiwan. Water 2021, 13, 2090. https://doi.org/10.3390/w13152090

- Bathrellos, G.D.; Skilodimou, H.D. Land Use Planning for Natural Hazards. Land 2019, 8, 128. https://doi.org/10.3390/land8090128

[Answer 8]:

Thanks for your good and positive comments. Your suggestion should be adopted. In the text we have added the related literature of the journal

  1. Bathrellos, G.D.; Skilodimou, H.D. Land use planning for Natural Hazards. Land. 2019, 8, 128-131. https://doi.org/10.3390/land8090128
  2. Trieu,; Bergström, P.; Sjödahl, M.; Hellström, J. G. I.; Andreasson, P.; Lycksam, H. Photogrammetry for Free Surface Flow Velocity Measurement: From Laboratory to Field Measurements. Water. 2021,13,1675-1688. https://doi.org/10.3390/w13121675.
  3. He, Y,;Sun,R,;Xu, Z,; Tang,  The Dynamic Change and Effect of Rainfall Induced Groundwater Flow. Water. 2021, 13, 2625-2638. https://doi.org/10.3390/w13192625.
  4. Wu, C.; Lin, C. Spatiotemporal Hotspots and Decadal Evolution of Extreme Rainfall-Induced Landslides: Case Studies in Southern Taiwan. Water.2021, 13, 2090-2108. https://doi.org/10.3390/w13152090

Except for the reviewers’ comments, we also made some changes in the revised manuscript. These changes will not influence the content and framework of the paper. And here we did not list the changes but marked in red in revised paper.

We appreciate for Editors/Reviewer’s warm work earnestly, and hope that the corrections will meet with approval. Please feel free to contact us with any questions and we are looking forward to your consideration.

Once again, thank you very much for your comments and suggestions.

Best regards,

Tianfeng Gu, Jiading Wang, Herry Lin, Qiang Xue, Bin Sun, Jiaxu Kong, Jiaxing Sun, Chenxing Wang, Fanchen Zhang.  

Reviewer 4 Report

The paper illustrates the characteristics, the factors, the causes and the evolution of landslides in an area of the NW of China, affected by the presence of a cover of loess on limestone bedrock.

The paper is well structured, exhaustive in defining the objectives and the results obtained from the analysis conducted.

Only two observations:

1) in Fig. 1 it is not clear the position of the piezometric surface, which is not reported in the geological section. It even seems that the aquifers are 2, one with a free piezometric surface (upper) and one under pressure, below the level of clay.

This hydrogeological situation deserves to be better explained, due to its possible effects on the landslide susceptib ility of the area.

2) the text can be significantly improved with a revision by an English native speaker.

Author Response

Manuscript ID: water-1359995

Manuscript Title:   On the spatiotemporal relationship between landslides and mechanisms on the Heifangtai terrace, Northwest China

Dear Editor,

We would like to thank you for sending our work for review, and the reviewers for their insightful and constructive comments. We have revised our manuscript, as highlighted in red color in the main body according to comments of the reviewers. We hope that this revised version will satisfy the reviewers.

The point by point responds to the reviewer’s comments are as follows:

[Question 1]

1) in Fig. 1 it is not clear the position of the piezometric surface, which is not reported in the geological section. It even seems that the aquifers are 2, one with a free piezometric surface (upper) and one under pressure, below the level of clay.

This hydrogeological situation deserves to be better explained, due to its possible effects on the landslide susceptib ility of the area.

[Answer 1]:

Thanks for your good and positive comments. Your suggestion should be adopted.

In Figure 1, we have marked the position of the piezometric surface, the exact location of which can be seen in Figure 1.

Figure 1. a. Study area, b. Interpretative diagram of lithological profile of Dangchuan landslide cluster and site monitoring layout.

[Question 2]

2) The text can be significantly improved with a revision by an English native speake.

[Answer 2]:

In order to improve the text, we have sought revisions from native English speakers.

Except for the reviewers’ comments, we also made some changes in the revised manuscript. These changes will not influence the content and framework of the paper. And here we did not list the changes but marked in red in revised paper.

We appreciate for Editors/Reviewer’s warm work earnestly, and hope that the corrections will meet with approval. Please feel free to contact us with any questions and we are looking forward to your consideration.

Once again, thank you very much for your comments and suggestions.

Best regards,

Tianfeng Gu, Jiading Wang, Herry Lin, Qiang Xue, Bin Sun, Jiaxu Kong, Jiaxing Sun, Chenxing Wang, Fanchen Zhang.  

Round 2

Reviewer 2 Report

This manuscript presents an improved and good work.

The manuscript should be acceptable for publication in this form.

Reviewer 3 Report

The manuscript entitled “On the spatiotemporal relationship between landslides and mechanisms on the Heifangtai terrace, Northwest China”, by T. Gu, J. Wang, H. Lin, Q. Xue, B. Sun, J. Kong, J. Sun, C. Wang, F. Zhang, and X. Wang, presents an improved and good work.

The manuscript should be acceptable for publication in the present form.